



# East Greenland Ice Sheet retreat history during the last deglaciation

Jacob T.H. Anderson[1], Nicolás E. Young[1], Allie Balter-Kennedy[1], Karlee K. Prince[2], Caleb K. Walcott-George[2], Brandon L. Graham[3], Joanna Charton[1], Jason P. Briner[2], and Joerg M. Schaefer[1]

[1]Lamont-Doherty Earth Observatory, Columbia University, Palisades, NY, USA

[2]Department of Earth Sciences, University at Buffalo, Buffalo, NY, USA

[3]United States Geological Survey, Florence Bascom Geoscience Center, Reston, VA, USA

Correspondence to: Jacob Anderson (janderson@ldeo.columbia.edu)

**Abstract**

The lack of geological constraints on past ice-sheet change in marine-based sectors of the Greenland Ice Sheet (GrIS) following the Last Glacial Maximum limits our ability to assess (1) the drivers of ice-sheet change, and (2) the performance of ice-sheet models that are benchmarked against the paleo-record of GrIS change. Here, we provide new in situ [10]Be surface exposure chronologies of ice-sheet margin retreat from the outer Scoresby Sund and Storstrømmen Glacier regions in eastern and northeastern Greenland,

respectively. Ice retreated from Rathbone Island, east of Scoresby Sund, by ~14.1 ka, recording some of the earliest documentations of terrestrial deglaciation in Greenland. The mouth of Scoresby Sund deglaciated by ~13.2 ka, and retreated at an average rate of ~43 m/yr between 13.2 ka and 9.7 ka. Storstrømmen Glacier retreated from the outer coast to within ~3 km of the modern ice margin between ~12.7 ka and 8.6 ka at an average rate of ~28 m/yr. Retreat then slowed or reached a stillstand as ice

retreated ~3 km between ~8.6 ka to the modern ice margin at ~8.0 ka. These retreat rates are consistent with late glacial and Holocene estimates for marine-terminating outlet glaciers across East Greenland, and comparable to modern retreat rates observed at the largest ice streams in northeastern, and northwestern Greenland.

## 25    1 Introduction

The rate of Greenland Ice Sheet (GrIS) mass loss and contribution to sea-level rise has increased over recent decades (Khan et al., 2022; The IMBIE Team, 2020). Projections under varying warming scenarios suggest that GrIS contributions to global mean sea level by 2100 range from -0.02–0.15 m to 0.05–0.23 m under SSP1-2.6 and SSP5-8.5 scenarios, respectively (IPCC, 2021). To place these contemporary short-term ice



sheet dynamics in context, we examine retreat patterns of the GrIS since the Last Glacial Maximum (LGM;
      26.5 – 19 ka; Clark et al., 2009).

      Previous research has provided robust constraints on the timing and pattern of deglaciation in southwestern
      Greenland (e.g., Briner et al., 2020; Larsen et al., 2014; Lesnek et al., 2020; Levy et al., 2012; Young et al.,
      2011, 2020, 2021). Relying partly on these detailed constraints, Briner et al. (2020) suggested rates of

southwestern GrIS mass loss, and perhaps the entire GrIS, over the twenty-first century are predicted to
      exceed rates experienced anytime in the last 12,000 years. Comparable geologic records from eastern (e.g.,
      Larsen et al., 2020; 2022; Levy et al., 2016; Roberts et al., 2024) and fewer studies from northern (e.g.
      Kelly & Bennike, 1992; O'Regan et al., 2021) sectors of Greenland, have been used to constrain the timing
      and rates of ice sheet margin retreat during the last deglaciation. In some regions across East and North

Greenland, however, data gaps hinder our ability to reconstruct the pattern and timing of GrIS retreat, and
      assess the accuracy of ice sheet models. This is particularly important in northeast Greenland, where major
      marine-terminating ice streams drain large portions of the GrIS. For example, the Northeast Greenland Ice
      Stream (NEGIS) is one of the largest drainage systems in Greenland, accounting for ~16% of the total GrIS
      discharge through three major marine-terminating outlet glaciers: Nioghalvfjerdsfjord Glacier (or 79N

Glacier), Zachariae Isstrøm (ZI), and Storstrømmen Isstrom (SI)(Hvidberg et al., 2020). Collectively, these
      glaciers contain 1.4 m of sea level equivalent (An et al., 2021; Rignot et al., 2022).

      During the LGM, the East GrIS extended onto the continental shelf edge, with significant contributions of
      ice from expanded marine terminating ice streams such as NEGIS and in the Scoresby Sund region (Fig.
      1). Marine geological data and geophysical surveys identifying glacial lineations and recessional moraines

have been used to reconstruct patterns of ice sheet retreat along the continental shelf during the last
      deglaciation (Arndt et al., 2015, 2017; Evans et al., 2009; López-Quirós et al., 2024; Ó Cofaigh et al., 2004,
      2025; Stein et al., 1996). On the northeast Greenland shelf, radiocarbon dates from marine sediments show
      that ice sheet retreat was underway on the shelf edge by 21.6 cal ka BP at Norske Trough, and by 19 cal
      BP at Westwind Trough (Ó Cofaigh et al., 2025). Ice retreated to the mid-shelf before 18.6 cal ka BP (Ó

Cofaigh et al., 2025), and that ice retreated from the inner continental shelf east of NEGIS between 13.4 –
      12.5 cal ka BP, which was primarily driven by an influx of relatively warm Atlantic water (Davies et al.,
      2022; Hansen et al., 2022).

      Despite recent advances understanding the retreat history from the outer coastline to the modern ice margin
      of marine-terminating glaciers along eastern Greenland (e.g., Håkansson et al., 2007; Larsen et al., 2020,

2022; Leger et al., 2024; Levy et al., 2016a), there are still major gaps in some regions (i.e., a lack of
      consistent ages or an absence of dated material). Here, we build on previous work studying the retreat
      history of marine-terminating glaciers in East Greenland, and present in situ cosmogenic [10]Be exposure



ages from the outer coast of Scoresby Sund, and near the modern ice margin of Storstrømmen Glacier to constrain East GrIS retreat. We integrate these data to calculate rates of ice margin retreat, and reconstruct a detailed East GrIS retreat pattern that we connect to potential mechanisms driving these GrIS changes.





**Figure 1.** Study area of East Greenland. The maximum LGM extent is marked by the black dashed line, derived from Leger et al. (2024 and references therein). The ice margin at ~11 ka and 9 ka is marked by the red and green dashed lines, derived from Larsen et al. (2020). Previously reported $^{10}$Be ages from outer coast, intermediate and ice margin sites (Dyke et al., 2014; Håkansson et al., 2007; Hughes et al., 2012; Larsen et al., 2022; Levy et al., 2016). Radiocarbon ages are listed in Table 1. The black rectangles show the locations of Scoresby Sund (Fig. 3a) and Dove Bugt region (Fig. 4). Basemap derived from IBCAO bathymetry (Jakobsson et al., 2024).





**Figure 2.** Representative boulder sample sites from outer Scoresby Sund (a, b, c), and the unnamed island near the terminus of Storstrømmen Glacier (d, e, f). (a) Sample 22-GRO-101, Uunarteq at the northern mouth of Scoresby Sund. (b) Sample 22GRO-131, Kap Brewster (Kangikajiip Appalia) at the southern mouth of Scoresby Sund. (c) Sample 22GRO-142, Rathbone Island, ~3 km to the east of the main coastline of Liverpool Land. (d) Sample 23DMH-29, from the prominent M1 moraine on the island at the terminus





of Storstrømmen Glacier. (e) Sample 23DMH-33, adjacent to the historical moraine and modern ice margin. (f) Sample 23DMH-20, outboard of the moraine.

## 2 Study sites and geological setting

The eastern margin of Greenland, from 79 N Glacier in the north, to Scoresby Sund in the south, comprises an extensive system of marine-terminating outlet glaciers, ice-free valleys, and fjords at lower elevations that transect blockfield-covered plateaus at higher elevations (Skov et al., 2020). Retreat of the ice margin along East Greenland has been constrained by marine and terrestrial radiocarbon ages (Table 1) and surface exposure ages (Fig. 1). Along the eastern margin of Greenland, the ice-free landscapes comprise moraines,

erratic boulders and polished bedrock surfaces that can be directly dated via cosmogenic nuclide exposure dating (e.g., Larsen et al., 2022; Levy et al., 2016; Skov et al., 2020). Grounded ice between Peary Land, 79 N Glacier, and ZI retreated westward out of the Greenland Sea and reached the present-day outer coastline by ~11.4 ka (Larsen et al., 2020). Between Storstrømmen Glacier and Kong Oscar Fjord, the GrIS receded to the outer coast between ~12.8 ka and 11.5 ka (Larsen et al., 2022). The ice sheet subsequently

retreated from the outer coast toward intermediate sites on the northeastern margin between ~11 – 9 ka (Larsen et al., 2018, 2022). The 79N Glacier rapidly deglaciated between 10 and 8.5 ka (Roberts et al., 2024). Ice reached close to the modern ice margin by ~9 ka at Lambert Land and had retreated to Borgfjorden by ~8.9 ka in the Dove Bugt region, and Vandrepasset by ~8.6 ka in Bessel Fjord (Larsen et al., 2018, 2022).

In southeastern Greenland, deglaciation of the ice sheet began at ~16 cal ka BP (Funder et al., 2011; Kuijpers et al., 2003), and receded to the mid shelf by ~14.7 cal ka BP at the Kangerlussuaq Trough (Jennings et al., 2006) and near Bernstorffs Fjord (Kuijpers et al., 2003). Marine-terminating outlet glaciers retreated to the present-day outer coastline by ~11.8 ka at Kangerlussuaq Fjord (Dyke et al., 2014), by 10.7 ka at Helheim Glacier (Hughes et al., 2012), by ~10.3 ka at Bernstorffs Fjord (Dyke et al., 2014), and by

11.3 ka at Skjoldungen (Levy et al., 2020).

A combination of raised marine beaches and terrestrial geological evidence has provided retreat chronologies of GrIS from the outer coast to the present-day ice margin (Fig. 1, Leger et al., 2024). Geomorphological mapping and dating glacial landforms and deposits using terrestrial cosmogenic nuclides (TCN; in situ $^{10}$Be and $^{14}$C) and radiocarbon ages have provided direct constraints of ice margin

retreat (e.g. Håkansson et al., 2007; Larsen et al., 2022; Levy et al., 2016). Our study sites focus on the glacial geology of two marine embayments: Scoresby Sund and Dove Bugt (Figs. 1, 2).



### 2.1 Scoresby Sund

Scoresby Sund (Kangertittivaq) is a large (13,700 km$^2$) fjord system situated between Jameson Land and
Liverpool Land in the north, King Christian IX Land in the south, and Milne Land, and the Renland
Peninsula in the west (Fig. 3). The fjord system extends ~350 km inland from the outer coast (Dowdeswell
et al., 1994a), with the outer fjord being ~120 km long, up to 50 km wide, and up to 650 m deep. Scoresby
Sund is deepest in the south and gently rises to the north towards Jameson Land (Håkansson et al., 2007).
At the fjord mouth is a large submarine moraine, Kap Brewster moraine (Fig. 3b; Dowdeswell et al., 1994;
Funder & Hansen, 1996).

Our study focuses on three presently ice-free sites on the outer coast of Scoresby Sund: Uunarteq on the
northern mouth, Kap Brewster (Kangikajiip Appalia) on the southern mouth, and Rathbone Island, ~3 km
to the east of the eastern coastline of Liverpool Land. At Uunarteq on the northern cape of Scoresby Sund,
the local bedrock comprises Paleoproterozoic orthogneiss and granite (Kokfelt et al., 2023). Erratic
boulders and cobbles of orthogneiss form a drift sheet over the local bedrock (Fig. 2a). At Kap Brewster on
the southern cape of Scoresby Sund, the local bedrock comprises Paleogene basalts of the North Atlantic
Igneous Province (Wager, 1947). The eastern margin of Kap Brewster is characterized by steep basalt cliffs
that extend over 200 m asl. Erratic sandstone and orthogneiss boulders and cobbles (>230 m asl) form a
drift draped over the basaltic plateau above the cliffs (Fig 2b; Håkansson et al., 2007). At Rathbone Island,
a thin drift of orthogneiss boulders and cobbles to at least 120 m asl overlie Paleozoic granodiorite and
quartz diorite bedrock (Kokfelt et al., 2023; Fig 2c).

Surface exposure $^{10}$Be ages from the mouth of Scoresby Sund suggest the ice margin retreated west of Kap
Brewster by ~14.5 ka (Håkansson et al., 2007). In Milne Land, ~150 km west of Kap Brewster, $^{10}$Be
exposure ages from the Milne Land Stade moraines suggest ice retreated to southeastern Milne Land by
~11.4 ka (Levy et al., 2016). Notably, the Milne Land stade moraines have been mapped from Milne Land
to Germania Land, and $^{10}$Be and radiocarbon ages constrain both local mountain glacier and GrIS retreat
during the Younger Dryas (12.9 – 11.7 ka; Funder et al., 2011; Kelly et al., 2008; Levy et al., 2016). In
northwestern Scoresby Sund, the Milne Land stade moraines grade into raised marine terraces (Funder,
1978), and radiocarbon dates from Kjove Land, suggested Milne Land stade moraines were deposited
before 12.4 cal ka BP (Hall et al., 2008). Additionally, radiocarbon dated shell samples from Schuchert Dal
reveal the Milne Land stade moraines were deposited during the early Younger Dryas or late Allerød (Hall
et al., 2010). The surface exposure ages in Scoresby Sund generally agree with radiocarbon ages from
marine sediment cores (Dowdeswell et al., 1994; Marienfeld, 1991), bivalves (Funder, 1978, 1990; Hall et





al., 2008), and plant macrofossils (Bennike et al., 1999; Hansen, 2001; Ingólfsson et al., 1994) which record

ice retreat through the main trough of Scoresby Sund between 11.7 and 7.9 cal ka BP (Fig. 3a).







**Figure 3.** (A) Scoresby Sund with new measured ¹⁰Be ages of boulders (red circles) with red sample names, and recalculated ¹⁰Be ages of Håkansson et al. (2007) with blue sample names at Kap Brewster. All ¹⁰Be ages shown in kiloyears with 1σ internal uncertainties. Mean ¹⁰Be age (green circle) from Milne Land (Levy et al., 2016). Radiocarbon ages (orange circles) are listed in Table 1. ¹⁰Be age outliers shown in italics. The grey dashed line denotes the flowline used to calculate average retreat rates. The maximum LGM extent is marked by the black dashed line, derived from Leger et al. (2024 and references therein). (B) East Scoresby Sund region showing the timing of ice margin retreat at the mouth of Scoresby Sund. The extent of the Kap Brewster submarine moraine is from Funder and Hansen (1996).

### 2.2 Storstrømmen Isstrom

Storstrømmen Isstrom is a major NEGIS outlet, and includes Kofoed-Hansen Bræ which drains to the northeast, and Storstrømmen Glacier which drains to the southeast (Fig. 4). The drainage basin (58,176 km²) of Storstrømmen Glacier contains enough ice to raise global sea level by 27 cm (Rignot et al., 2022). The southeastern lobe of Storstrømmen Isstrom is a 23 km wide marine-terminating outlet glacier that drains into Borgfjorden of the Dove Bugt Embayment (Fig. 4). Storstrømmen is a surge glacier, with an estimated recurrence cycle of 50 – 70 years (Mouginot et al., 2018; Reeh et al., 1994). The terminus retreated 15 km between 1913 and 1950, remained stable between 1950 and 1978, and readvanced 8 km between 1978 and 1984 during a surge (Reeh et al., 1994). Grounding line retreat of 1.1 km from 2017 – 2021 has been attributed to increased surface ablation (Rignot et al., 2022). Recent observations suggest a possible surge onset projected to occur between 2027 and 2040 (Andersen et al., 2025).

At the terminus, Storstrømmen Glacier splits into two lobes that flow around an unnamed island where we collected samples for surface exposure dating. The western lobe merges with L. Bistrup Brae Glacier, and the eastern lobe flows directly into Borgfjorden. The unnamed island acts as a pinning point, stabilizing the present-day glacier (Reeh et al., 1994). The local bedrock of the island comprises basement Paleoproterozoic orthogneiss (Kokfelt et al., 2023) and is overlain by moraines and glacially transported erratics (Fig 2d,e,f). At the southern end of the island, moraine deposits grade into a raised marine terrace at the local marine limit (28 m asl). This observation is consistent with the 28 m marine limit at Hellefjord in western Germania Land (Landvik, 1994; Weidick et al., 1996). Radiocarbon ages from marine bivalves in Neoglacial moraines indicate that Storstrømmen Glacier terminated inland from its present-day position between 5.4 – 1.2 cal ka BP, when a ~100 km long sound formed between Kofoed-Hansen Bræ and Storstrømmen (Weidick et al., 1996). The ice margin of the unnamed island is surrounded by fresh (little weathered) "historical moraine" ridges and mark a late Neoglacial readvance that culminated ~1850 (Weidick et al., 1996).





**Table 1.** Radiocarbon ages from eastern Greenland

| Sample number | Latitude | Longitude | Location | Material dated | Radiocarbon age (¹⁴C yr BP) | Radiocarbon age uncertainty (1σ) | Calibrated age (cal yr. BP, median) | Calibrated age range (cal yr. BP, 2σ) | Reference |
|---|---|---|---|---|---|---|---|---|---|
| **Northeast Greenland shelf** | | | | | | | | | |
| ETH-106030 | 80.037 | -8.923 | DA17-NG-ST03-039G | Mixed planktonic foraminifera | 12070 | 60 | 13291* | 13202-13584* | Hansen et al. (2022) |
| ETH-113888 | 78.501 | -16.28 | DA17-NG-ST08-092G | Mixed benthic foraminifera | 10200 | 190 | 12473* | 11268-12595* | Davies et al. (2022) |
| AWI 2795.1.1 | 77.497 | -18.14 | PS100/270 | Mixed benthic foraminifera | 9437 | 104 | 10138 | 9670-10568 | Syring et al. (2020) |
| 121063.1.1 | 77.127 | -10.68 | DA17-NG-ST12–135G | Mixed benthic foraminifera | 14450 | 220 | 16684 | 16080-17285 | Lopez-Quiros et al. (2024) |
| **Lambert Land** | | | | | | | | | |
| 442909 | 79.107 | -19.75 | Zachariae Isstrom | Marine bivalve | 7595 | 55 | 7905 | 7624-8184 | Bennike and Weidick (2001) |
| **Clavering Ø** | | | | | | | | | |
| I-9659 | 74.350 | -21.87 | Clavering Ø | Marine bivalve | 8195 | 125 | 8148 | 7050-9396 | Weidick (1978) |
| **Kejser Franz Joseph Fjord** | | | | | | | | | |
| MM/G3 | 73.550 | -25.12 | Kejser Franz Joseph Fjord | Marine bivalve | 8020 | 100 | 8078 | 7179-9086 | Wagner and Melles (2002) |
| 60282.1.1 | 73.675167 | -24.18 | HH13-012 | Mixed benthic foram | 7470 | 130 | 7471 | 6518-8376 | Olsen et al. (2022) |
| 4491.1.1 | 73.474667 | -24.61 | PS2633 | Foraminifera | 8376 | 84 | 8452 | 7550-9447 | Olsen et al. (2022) |
| PS2630 | 73.158 | -18.07 | PS2630 | Foraminifera | 13560 | 130 | 15544 | 15113-15989 | Evans et al. (2002) |
| **Kong Oscar Fjord** | | | | | | | | | |
| I-9104 | 73.350 | -26.47 | Kong Oscar Fjord | Marine bivalve | 7760 | 115 | 7985 | 7654-8329 | Weidick (1977) |
| I-9658 | 73.167 | -27.42 | Kong Oscar Fjord | Marine bivalve | 7160 | 125 | 7428 | 7067-7779 | Weidick (1978) |
| Lu-1070 | 72.717 | -26.83 | Kong Oscar Fjord | Marine bivalve | 7530 | 75 | 7698 | 7150-8297 | Håkansson (1976) |
| **Scoresby Sund** | | | | | | | | | |
| UtC-7428 | 70.595 | -21.54 | Scoresby Sund | Plant macrofossil | 8900 | 50 | 10026 | 9887-10197 | Cremer et al. (2001) |
| AAR-199 | 70.338 | -23.71 | PS1715-1 | Foraminifera | 10340 | 150 | 11250 | 10674-11819 | Marienfeld (1991) |



| Sample ID | Latitude | $\delta^{13}C$ | Location | Material | $^{14}C$ age | error | Cal. age | Cal. range | Reference |
|---|---|---|---|---|---|---|---|---|---|
| AAR-200 | 70.483 | -24.67 | PS1719-1 | Foraminifera | 10480 | 190 | 11645 | 11058 - 12392 | Marienfeld (1991) |
| AAR-202 | 70.708 | -25.00 | PS1728-1 | Foraminifera | 10760 | 120 | 12023 | 11503 - 12486 | Marienfeld (1991) |
| AAR-201 | 70.928 | -24.98 | PS1727-1 | Foraminifera | 9080 | 650 | 9589 | 7952 - 11272 | Marienfeld (1991); Dowdeswell et al. (1994) |
| K-1461 | 71.167 | -25.33 | Scoresby Sund | Marine bivalve | 9040 | 140 | 9650 | 9275 - 10108 | Funder (1990) |
| K-3109 | 70.500 | -23.45 | Scoresby Sund | Marine bivalve | 10560 | 145 | 11726 | 11199 - 12318 | Funder (1990) |
| AAR-636 | 70.533 | -23.63 | Scoresby Sund | Marine bivalve | 9170 | 140 | 9764 | 9398 - 10183 | Gulliksen et al. (1991) |
| I-9492 | 70.500 | -26.18 | Scoresby Sund | Marine bivalve | 9925 | 140 | 10882 | 10395 - 11310 | Funder (1978) |
| AAR-3368 | 70.387 | -23.87 | Scoresby Sund | Plant macrofossils | 9780 | 160 | 11186 | 10695 - 11743 | Hansen (2001) |
| 100380, 100381 | 70.843 | -24.01 | Scoresby Sund | Plant macrofossils | 9665 | 55 | 11058 | 10780 - 11207 | Bennike et al. (1999) |
| AAR-2540 | 70.850 | -24.02 | Scoresby Sund | Plant macrofossils | 10130 | 130 | 11729 | 11257 - 12177 | Bennike et al. (1999) |
| AAR-3369 | 70.850 | -24.07 | Scoresby Sund | Plant macrofossils | 9440 | 65 | 10681 | 10501 - 11069 | Hansen (2001) |
| AAR-3802 | 70.850 | -24.15 | Scoresby Sund | Marine bivalve | 9985 | 60 | 10915 | 10581 - 11203 | Hansen (2001) |
| AAR-2571 | 70.985 | -24.00 | Scoresby Sund | Plant macrofossils | 7900 | 100 | 8751 | 8464 - 9006 | Böcher and Bennike (1996) |
| AAR-1829 | 70.959444 | -24.11361 | Scoresby Sund | Marine bivalve | 10140 | 100 | 11126 | 10696 - 11557 | Funder and Hansen (1996) |
| AAR-2541 | 70.950 | -24.12 | Scoresby Sund | Plant macrofossils | 9910 | 130 | 11411 | 10885 - 11873 | Hansen (2001) |
| AAR-644 | 70.914 | -24.21 | Scoresby Sund | Marine bivalve | 9670 | 180 | 10486 | 9930 - 11085 | Ingolfsson et al. (1994) |
| 87516 | 71.567 | -23.97 | Scoresby Sund | Plant macrofossils | 7000 | 390 | 7848 | 7003 - 8601 | Böcher and Bennike (1996); Bennike et al. (1999) |
| K-1915 | 71.350 | -24.83 | Scoresby Sund | Marine bivalve | 10300 | 120 | 11423 | 11018 - 11902 | Funder (1978) |
| I-5421 | 70.950 | -28.15 | Scoresby Sund | Marine bivalve | 7540 | 130 | 7863 | 7560 - 8182 | Funder (1978) |
| **Kangerlussuaq Trough** | | | | | | | | | |
| AA-4026 | 66.203 | -30.66 | 10A | Foraminifera | 13585 | 110 | 15628 | 15259 - 15995 | Williams (1993) |
| AA-6848 | 66.764 | -30.84 | PO-175/15 | Foraminifera | 14845 | 190 | 17229 | 16717 - 17791 | Williams et al. (1995) |





180

| | | | | | | | | | |
|---|---|---|---|---|---|---|---|---|---|
| AA-11584 | 68.116 | -31.43 | 91-K11 A | Foraminifera | 9975 | 100 | 10942 | 10585 - 11239 | Andrews et al. (1997) |
| CAMS-32047 | 68.100 | -29.35 | JM96-1207 | Foraminifera | 9800 | 60 | 10731 | 10495 - 11037 | Jennings et al. (2002) |
| AA-29205 | 67.300 | -30.97 | JM96-1214/2GC | Benthic foraminifera | 11,380 | 80 | 12786 | 12601 - 13009 | Smith and Licht (2000); Jennings et al. (2006) |
| AA-32045 | 67.047 | -30.86 | JM96-1215/2GC | Benthic and planktic foraminifera | 12900 | 50 | 14693 | 14326 - 15010 | Smith and Licht (2000); Jennings et al (2006) |
| AA-23221 | 65.963 | -30.63 | JM96-1216/2-GC | Benthic foraminifera | 14550 | 150 | 16862 | 16423 - 17296 | Smith and Licht (2000); Jennings et al. (2008) |
| AA-43116 | 68.093 | -27.84 | MD99-2317 | Bryozoans | 11950 | 110 | 13405 | 13122 - 13717 | Jennings et al. (2006) |
| AA-4666 | 67.410 | -31.07 | 3 | Foraminifera | 9375 | 70 | 10133 | 9818 - 10424 | Williams (1993); Mienert et al. (1992) |

*Modelled median age derived from age-depth model (Davies et al., 2022; Hansen et al., 2022).





**Figure 4.** (A) Dove Bugt region showing outer coast (purple), intermediate (green) and ice margin (blue)

exposure ages from Larsen et al. (2022). Black inset shows the location of Storstrømmen Glacier terminus



in aerial photo (B) and in Fig. 5. The grey dashed line denotes the flowline used to calculate average retreat rates. (B) View looking south at the Storstrømmen Glacier terminus showing the historical moraine (blue dashed line) and modern ice margin.

## 3 Methods

### 3.1 Field methods

We collected samples from the mouth of Scoresby Sund and Rathbone Island during the boreal summer of 2022, and from the island at the terminus of Storstrømmen Glacier during the boreal summer of 2023. We sampled 13 boulders from Scoresby Sund (Fig. 3), and 15 boulders and one bedrock surface from Storstrømmen (Fig. 5) for $^{10}$Be analysis (Table 2). We targeted large (>0.5 m diameter) boulders perched on bedrock or on well-developed moraines. The bedrock sample (23DMH-CR1-SURFACE) was collected from a surface with visible signs of glacial abrasion (e.g. glacial polish, striae). Four of the samples from Kap Brewster at the mouth of Scoresby Sund were collected from friable white sandstone boulders displaying exfoliation and weathering pits. We avoided sampling surfaces near the exfoliation or weathering pits. Two erratic boulder samples (22GRO-126 and 22GRO-127) from Kap Brewster were collected from the same boulders (KB-1 and KB-2) as previously sampled by Håkansson et al. (2007).

### 3.2 Analytical methods

Whole rock samples from Scoresby Sund were crushed, sieved and processed to quartz purification at the University at Buffalo. Whole rock samples from Storstrømmen Glacier were crushed, sieved and processed to quartz purification at Lamont-Doherty Earth Observatory. All samples were then processed at Lamont-Doherty Earth Observatory via conventional $^{10}$Be extraction methods (Schaefer et al., 2009). $^{10}$Be/$^9$Be ratios were measured at the Center for Accelerator Mass Spectrometry at Lawrence Livermore National Laboratory, and normalized to the 07KNSTD standard of Nishiizumi et al. (2007) with a nominal ratio of 2.85 x10$^{-12}$. Blank corrections to measured $^9$Be/$^{10}$Be ratios amounted to <2%, and process blanks range from 6200 – 11,855 $^{10}$Be atoms (Table S2). Final errors in $^{10}$Be concentrations are obtained by quadrature addition of the final accelerator mass spectrometry analytical error, and a 1% error in Be spike concentration.

We calculated our new surface exposure ages and recalculated previously reported ages using version 3 of the CRONUS-Earth calculator (http://hess.ess.washington.edu/, last access: February 2025; Balco et al., 2008) using the Lm scaling scheme (Lal, 1991; Stone, 2000) and the Arctic production rate (Young, et al., 2013a; Table S1). We note that using the St scaling scheme results in almost identical ages for our samples, while ages derived using the LSDn scaling scheme differ by less than 120 years, and thus do not affect our



interpretations. Four samples are quartz sandstone boulders collected from Kap Brewster (22GRO-126, 22GRO-127, 22GRO-128, 22GRO-131) and we use a density of 2.4 g cm$^{-3}$. All other samples are crystalline boulders or bedrock (e.g., orthogneiss, granodiorite) and we use a density of 2.65 g cm$^{-3}$. Unless otherwise

noted, all radiocarbon ages (reported as: cal ka BP) mentioned in the text are recalculated using CALIB 8.2 (Stuiver & Reimer, 1993); http://calib.org/calib/calib.html, last access: April 2025), and the IntCal20 calibration curve (Reimer et al., 2020) for terrestrial samples, or the Marine20 calibration curve (Heaton et al., 2020) for marine samples (Table 1, Table S3). For marine samples, marine reservoir corrections were calculated using the Marine Reservoir Correction Database (http://calib.org/marine/, last access: April

2025)(Reimer & Reimer, 2001). Each site uses a ΔR value based on water depth, distance to coast, and where possible species that are most representative of the study site, as suggested in Pearce et al. (2023).



**Table 2** Cosmogenic $^{10}$Be concentrations and apparent exposure ages.

| Sample name | Latitude (DD) | Longitude (DD) | Elevation (masl) | Sample thickness (cm) | Topographic shielding | $^{10}$Be conc. ($10^4$ atoms g$^{-1}$)[a] | $^{10}$Be age (ka)[b,c,d] |
|---|---|---|---|---|---|---|---|
| **Rathbone Island** | | | | | | | |
| 22GRO-139 | 70.66612 | -21.3985 | 118 | 3.00 | 1 | 7.66 ± 0.18 | 16.4 ± 0.4 (0.6) |
| 22GRO-140 | 70.66674 | -21.3991 | 119 | 3.00 | 1 | 8.25 ± 0.20 | 17.7 ± 0.4 (0.6) |
| 22GRO-142 | 70.66599 | -21.404 | 97 | 2.5 | 1 | 6.45 ± 0.16 | 14.1 ± 0.3 (0.5) |
| **Uunarteq** | | | | | | | |
| 22GRO-100 | 70.44798 | -21.8879 | 99 | 3.00 | 0.998 | 6.19 ± 0.15 | 13.6 ± 0.3 (0.5) |
| 22GRO-101 | 70.44798 | -21.8879 | 99 | 3.00 | 0.998 | 6.04 ± 0.17 | 13.2 ± 0.4 (0.5) |
| 22GRO-102 | 70.44791 | -21.8876 | 94 | 3.00 | 0.998 | 5.56 ± 0.23 | 12.3 ± 0.5 (0.6) |
| 22GRO-103 | 70.44786 | -21.888 | 91 | 3.00 | 0.994 | 5.64 ± 0.14 | 12.5 ± 0.3 (0.4) |
| 22GRO-104 | 70.44768 | -21.888 | 103 | 3.00 | 0.998 | 12.20 ± 0.26 | 26.4 ± 0.6 (0.9) |
| **Kap Brewster (Kangikajiip Appalia)** | | | | | | | |
| 22GRO-126 | 70.151 | -22.0754 | 225 | 3.00 | 1 | 5.99 ± 0.17 | *11.4 ± 0.3 (0.4)* |
| 22GRO-127 | 70.15112 | -22.076 | 231 | 3.00 | 1 | 6.97 ± 0.18 | 13.2 ± 0.3 (0.5) |
| 22GRO-128 | 70.15112 | -22.0761 | 233 | 3.00 | 1 | 7.18 ± 0.16 | 13.6 ± 0.3 (0.5) |
| 22GRO-131 | 70.15077 | -22.0766 | 227 | 3.00 | 1 | 7.33 ± 0.17 | 13.9 ± 0.3 (0.5) |
| 22GRO-134 | 70.15005 | -22.0756 | 230 | 3.00 | 1 | 7.03 ± 0.15 | 13.4 ± 0.3 (0.5) |
| **Weighted mean (Uunarteq and Kap Brewster):** | | | | | | | **13.2 ± 0.7 (n = 8)** |
| **Storstrømmen Glacier** | | | | | | | |
| 23DMH-CR1-SURFACE | 76.87522 | -22.3197 | 166 | 1.65 | 1 | 5.634 ± 0.15 | *11.5 ± 0.3 (0.4)* |
| 23DMH-18 | 76.87489 | -22.3197 | 161 | 2.79 | 1 | 5.93 ± 0.16 | *12.3 ± 0.3 (0.5)* |
| 23DMH-20 | 76.87555 | -22.3165 | 164 | 2.19 | 1 | 4.25 ± 0.15 | 8.7 ± 0.3 (0.4) |
| 23DMH-21 | 76.87409 | -22.3181 | 163 | 1.85 | 1 | 4.33 ± 0.09 | 8.9 ± 0.2 (0.3) |
| 23DMH-22 | 76.85698 | -22.277 | 114 | 1.08 | 1 | 4.48 ± 0.11 | *9.6 ± 0.2 (0.4)* |
| 23DMH-23 | 76.85442 | -22.281 | 92 | 3.53 | 1 | 3.82 ± 0.19 | 8.6 ± 0.4 (0.5) |

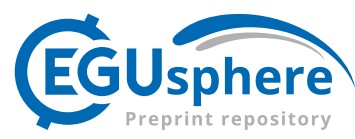

| Sample | | | | | | | |
|---|---|---|---|---|---|---|---|
| 23DMH-24 | 76.85425 | -22.2807 | 91 | 3.37 | 1 | 3.72 ± 0.08 | 8.4 ± 0.2 (0.3) |
| 23DMH-25 | 76.85428 | -22.2824 | 91 | 2.54 | 1 | 3.84 ± 0.12 | 8.6 ± 0.3 (0.3) |
| 23DMH-26 | 76.84808 | -22.2882 | 38 | 2.39 | 1 | 3.32 ± 0.13 | 7.9 ± 0.3 (0.4) |
| 23DMH-27 | 76.84876 | -22.2989 | 90 | 2.40 | 1 | 3.70 ± 0.09 | 8.3 ± 0.2 (0.3) |
| 23DMH-28 | 76.8511 | -22.3069 | 109 | 1.07 | 1 | 3.80 ± 0.09 | 8.2 ± 0.2 (0.3) |
| 23DMH-29 | 76.8687 | -22.3197 | 152 | 3.67 | 1 | 4.10 ± 0.11 | 8.6 ± 0.2 (0.3) |
| 23DMH-30 | 76.86914 | -22.3229 | 151 | 2.41 | 1 | 4.19 ± 0.11 | 8.7 ± 0.2 (0.3) |
| 23DMH-31 | 76.87427 | -22.3491 | 124 | 3.07 | 1 | 3.92 ± 0.09 | 8.5 ± 0.2 (0.3) |
| 23DMH-32 | 76.88953 | -22.3271 | 127 | 2.01 | 1 | 3.80 ± 0.10 | 8.1 ± 0.2 (0.3) |
| 23DMH-33 | 76.89405 | -22.3273 | 125 | 2.22 | 1 | 3.72 ± 0.09 | 8.0 ± 0.2 (0.3) |
| **Weighted mean:** | | | | | | | **8.4 ± 0.4 (n = 13)** |

[a] Samples normalized to the 07KNSTD standard of Nishiizumi et al. (2007).

[b] Exposure ages calculated using the CRONUS-Earth calculator (https://hess.ess.washington.edu/, last access: February 2025), using the Lm scaling scheme (Lal, 1991; Stone, 2000) and the Arctic production rate (Young, et al., 2013a).

[c] Both internal and external uncertainties shown at the 1σ level. Internal uncertainties are analytical uncertainties only, and external uncertainties (given in parentheses) are absolute uncertainties and include production rate and scaling errors.

[d] The weighted mean for each site includes the propagation of a 3.7% production rate uncertainty (Young et al., 2013a). Outliers in italics are excluded from the weighted mean age.



## 4 Results

### 4.1 Scoresby Sund

The 13 erratic boulder samples from the outer coast of the Scoresby Sund region produce [10]Be ages that range from $11.4 \pm 0.3 - 26.5 \pm 0.6$ ka (Figs. 3,6). On Rathbone Island, east of Scoresby Sund, three erratic boulders were sampled between 97 and 118 m elevation and produce exposure ages of $14.1 \pm 0.3$ ka (22GRO-142), $16.4 \pm 0.4$ ka (22GRO-139), and $17.7 \pm 0.4$ ka (22GRO-140). At Uunarteq located at the northern mouth of Scoresby Sund, five erratic boulders sampled between 91 and 103 m elevation produce [10]Be ages that range from $12.3 \pm 0.5 - 26.4 \pm 0.6$ ka. At Kap Brewster (Kangikajik) located at the southern mouth of Scoresby Sund, five erratic boulders sampled between 225 and 233 m elevation produce [10]Be ages that range from $11.4 \pm 0.3$ to $13.9 \pm 0.3$ ka. The [10]Be ages from samples 22GRO-126 and 22GRO-127 are $11.4 \pm 0.3$ ka, and $13.2 \pm 0.6$ ka, respectively; these samples were collected from the same boulders as samples KB-1 and KB-2 from Håkansson et al. (2007), which yield recalculated ages of $18.3 \pm 2.9$ and $17.2 \pm 2.2$ ka. The large difference in ages is outside the 1σ error range, and may be attributed to analytical errors. After excluding one older outlier ($26.4 \pm 0.6$ ka) from Uunarteq, and a younger outlier from Kap Brewster ($11.4 \pm 0.3$ ka), the mean age from the two sites at the mouth of Scoresby Sund is $13.2 \pm 0.7$ ka (n=8; Table 2).

### 4.2 Storstrømmen Glacier

The 15 erratic boulder samples from the unnamed island at the Storstrømmen Glacier terminus produce [10]Be ages that range from $7.9 \pm 0.3 - 12.3 \pm 0.3$ ka (Figs. 5,6). Excluding two outliers (23DMH-18, 23DMH-22) the mean age of the boulders on the island is $8.4 \pm 0.4$ ka (n = 13; Table 2). Of these, seven samples were collected from boulders on a prominent moraine (M1 moraine) deposited by the western lobe that extends ~5 km through the island, and produce [10]Be ages of $9.6 \pm 0.2$ ka, $8.6 \pm 0.4$ ka, $8.4 \pm 0.2$ ka, $8.6 \pm 0.3$ ka, $8.6 \pm 0.2$ ka, $8.7 \pm 0.2$ ka, and $8.5 \pm 0.2$ ka. The mean age for the M1 moraine is $8.6 \pm 0.3$ ka (n = 6), excluding sample 23DMH-22 ($9.6 \pm 0.2$ ka), which was identified as an outlier (Fig. 6). [10]Be ages from three boulders ~500 m north and outboard of the moraine are $8.7 \pm 0.3$, $8.9 \pm 0.2$, and $12.3 \pm 0.3$ ka. The $12.3 \pm 0.3$ ka sample (23DMH-18) outboard of the M1 moraine was identified as an outlier as it falls outside of the 2σ range of the mean age of the boulders on the island. The one bedrock surface, ~500 m north of the moraine and in close proximity to 23DMH-18, produced a [10]Be age of $11.5 \pm 0.3$ ka (23DMH-CR1-SURFACE). The bedrock surface likely has inherited [10]Be and was not used to determine deglacial history.



On the southeastern side of the island, $^{10}$Be ages from three boulders close to the raised marine terrace are 7.9 ± 0.3, 8.3 ± 0.2, and 8.2 ± 0.2 ka, are inboard of the M1 moraine. One the northern margin of the island, $^{10}$Be ages from two boulders adjacent to the historical moraine and modern ice margin are 8.1 ± 0.2 ka (23DMH-32) and 8.0 ± 0.2 ka (23DMH-33). These boulders were deposited by the eastern lobe.

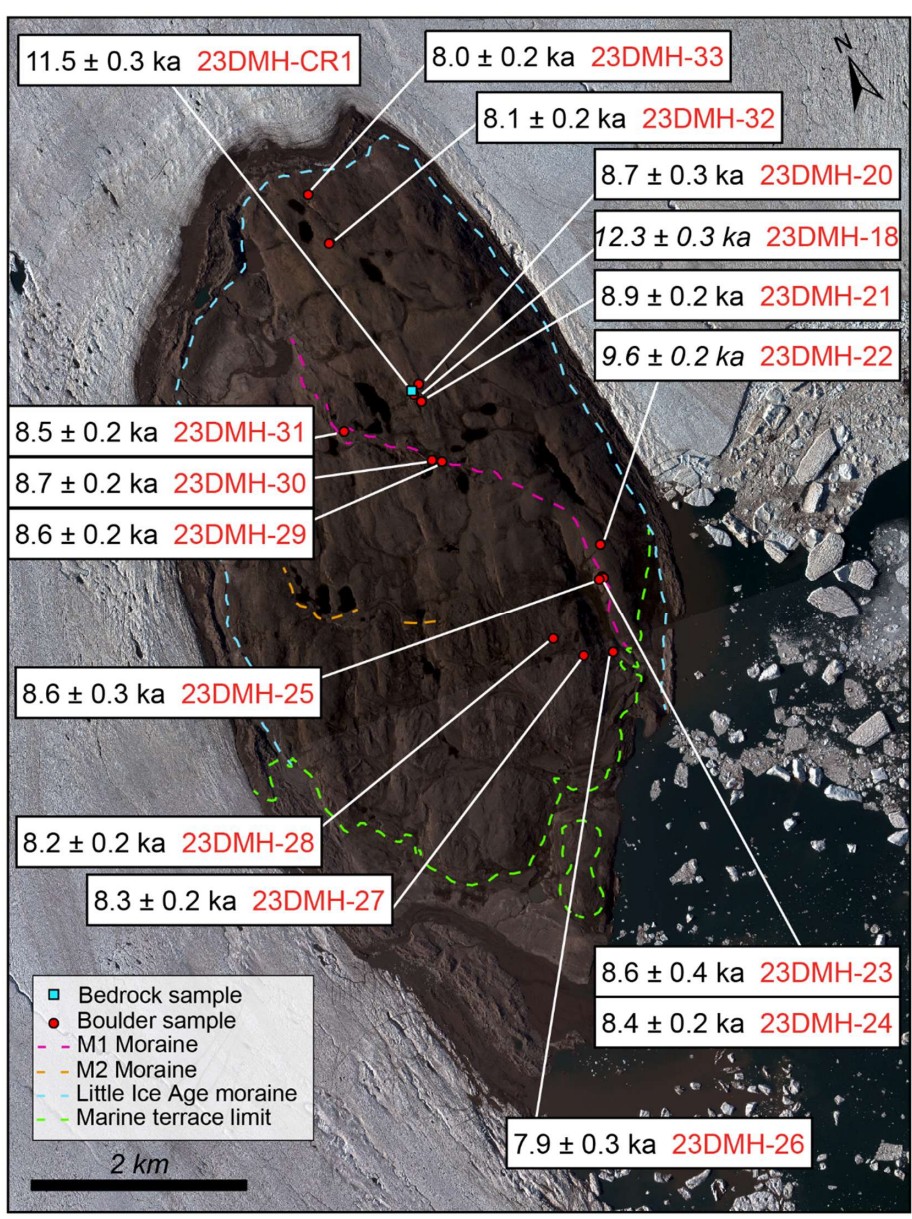



**Figure 5.** Map of Storstrømmen Glacier terminus with extent of Little Ice Age moraine (blue dashes) and raised marine terrace limit (green dashes). The M1 moraine (magenta) deposited by the western lobe, grades into the marine terrace at 28 m elevation. Measured $^{10}$Be ages of boulders (red circles) and bedrock (blue square) are shown in kiloyears (ka) with 1σ internal uncertainties. $^{10}$Be age outliers shown in italics. Satellite image from WorldView-2 (© 2021, Maxar).

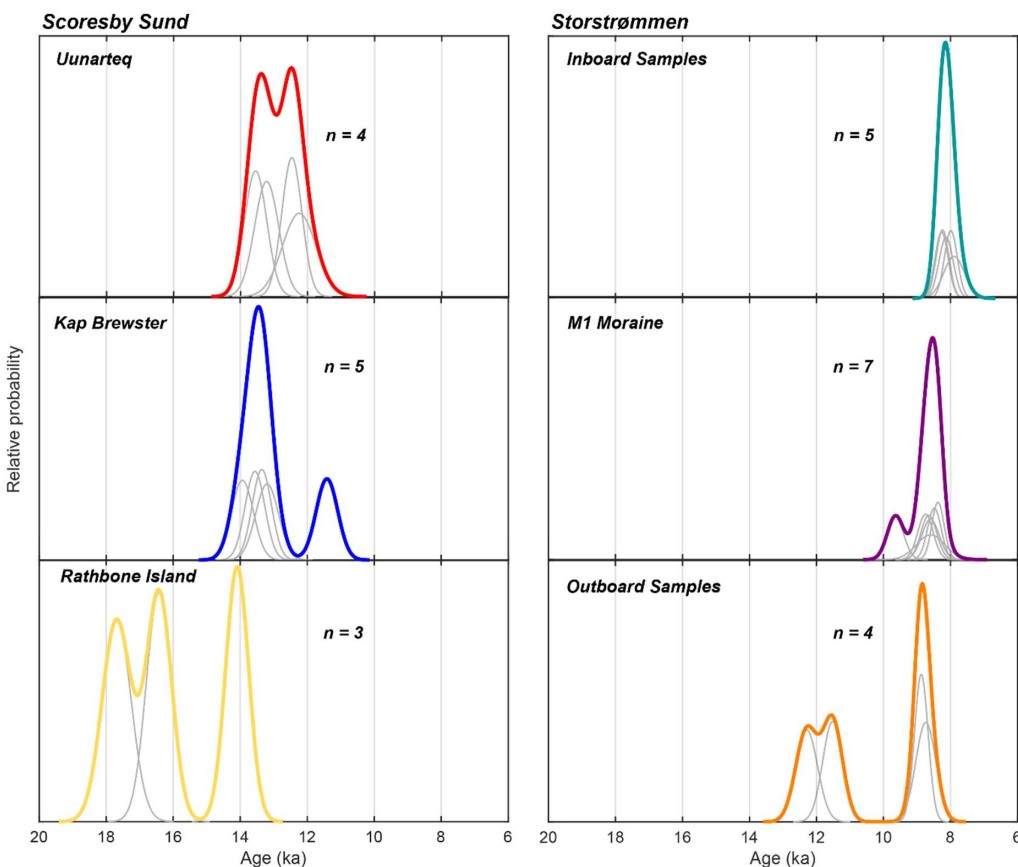

**Figure 6.** Camel plots (normal kernel density function) for $^{10}$Be exposure ages Uunarteq, Kap Brewster and Rathbone Island near the outer coast of Scoresby Sund, and samples from the M1 moraine, and inboard and outboard of the M1 moraine at the terminus of the Storstrømmen Glacier. Colored camel plots show the relative probability for all $^{10}$Be ages at each site.

## 5 Discussion

### 5.1 Glacier evolution in Scoresby Sund



The [10]Be exposure ages from Rathbone Island provide new minimum constraints on the retreat of the GrIS off the coast of Liverpool Land at 14.1 ± 0.3 ka, and record one of the earliest terrestrial deglaciation events in Greenland. Our mean [10]Be age of 13.2 ± 0.7 ka (n = 8) from the north and south sides of the mouth of Scoresby Sund is consistent with the minimum [10]Be exposure age from Kap Brewster reported by Håkansson et al. (2007) of 14.5 ± 1.7 ka, and provides a minimum constraint on ice margin retreat past the mouth of Scoresby Sund (Fig. 6). Based on our age constraints at the mouth of Scoresby Sund, we infer the depositional age of the Kap Brewster submarine moraine to correspond with our mean age of ~13.2 ka (Fig. 3b).

At sites (≥300 km) north of Scoresby Sund, the outer coast deglaciated between 12.8 ± 0.6 and 11.5 ± 0.2 ka (Larsen et al., 2022). We suggest the earlier onset of deglaciation at the outer coast of the Scoresby Sund region may be related to the closer proximity of our dated sites to the edge of the continental shelf than other parts of East Greenland. This interpretation is further supported by deglaciation chronologies before 13.4 cal ka BP, and 13.3 cal ka BP on the northeast Greenland shelf (Davies et al., 2022; Hansen et al., 2022).

The timing of retreat at Rathbone Island at ~14.1 ka, to the mouth of Scoresby Sund at ~13.2 ka, coincides with the Bølling-Allerød warm period (14.7 – 12.9 ka, Rasmussen et al., 2006), suggesting widespread retreat of the GrIS across the eastern continental shelf. Retreat from across the northeast Greenland shelf between 16.6 and 13.4 cal ka BP (Davies et al., 2022; López-Quirós et al., 2024), coincides broadly with Meltwater Pulse 1A, a period of rapid sea level rise between 14.6 and 14.3 ka (Deschamps et al., 2012; Hanebuth et al., 2000). Alongside our [10]Be ages from Rathbone Island, we suggest that ice margin retreat across the East Greenland shelf may have been influenced, in part, by enhanced sensitivity to abrupt global sea-level rise at this time. The retreat of ice in the Scoresby Sund region was likely attributed to the influx of warm Atlantic water that flowed onto the East Greenland continental shelf, perhaps related to the Bølling-Allerød warm period (Hansen et al., 2022). The inflow of Atlantic water beneath fresher and colder polar surface water has been inferred by high abundances of *Cassidulina neoteretis* (Foraminifera) in marine sediment cores, a species typically found in Atlantic sourced subsurface waters (Cage et al., 2021; Consolaro et al., 2018). In southeast Greenland, the high abundance of *C. neoteretis* in shelf cores also suggests the advection of Atlantic water into the Kangerlussuaq Trough drove basal melt and retreat during the Bølling-Allerød and the Younger Dryas (Jennings et al., 2006).

Our retreat chronology, and previously published [10]Be and radiocarbon ages in Scoresby Sund, suggest ice in the main fjord basin of Scoresby Sund retreated from the mouth of Scoresby Sund at ~13.2 ka to Renland by 9.7 cal ka BP (Funder, 1978), and using the flowline in Fig. 3a, we calculate an average retreat rate of

~43 m/yr. In southern Scoresby Sund, ice retreated ~150 km from the mouth to the southern end of Milne
       Land by ~11.4 ka (Levy et al., 2016) at an average retreat rate of ~83 m/yr.

## 5.2 Glacier evolution in the Dove Bugt region

       Our mean [10]Be age of 8.6 ± 0.3 ka (n = 6) from the prominent M1 moraine at Storstrømmen Glacier record
the conclusion of a stillstand or advance near the modern ice margin (Fig. 6). This age is consistent with
       the [10]Be retreat chronology at Bræ Øerne of 8.9 ± 0.2 ka (Larsen et al., 2022), ~14 km south of
       Storstrømmen Glacier (Fig. 4). We suggest Storstrømmen Glacier comprised two expanded ice lobes
       around the eastern and western margins of the unnamed island. As the glacier retreated, boulders and
       moraines were deposited between ~8.6 ka and 8.0 ka (Figs. 2,5). The western ice lobe deposited the
prominent M1 moraine in the middle of the island and all the boulders inboard (west) of the moraine. As
       the glacier retreated the western lobe wrapped around the southern end of the island depositing the M2
       moraine. Two lakes ~1.5 km south of sample 23DMH-31 on the M2 moraine are dammed at their southern
       end, and we interpret these to be moraine dammed lakes that formed as the western lobe retreated (Fig. 5).
       The eastern lobe ice likely retreated beyond the historical moraine, and present ice margin at ~8 ka.

The 28 m marine limits at the terminus of Storstrømmen (Weidick et al., 1996) is dated to an estimated age
       of ~8 cal ka BP with marine bivalves (Landvik, 1994; Fig. 7). Our exposure ages from the unnamed island
       are consistent with the radiocarbon ages of the raised marine sediments. Also, our ice margin retreat history
       is in agreement with deglaciation of inner Kejser Franz Joseph Fjord and Nordfjord, ~360 km south of
       Storstrømmen, constrained by radiocarbon ages of 8.5 and 7.5 cal ka BP, respectively (Olsen et al., 2022).

Our [10]Be ages from the M1 moraine at ~8.6 ka to near the modern ice margin ~8 ka suggests an ice margin
       stillstand or slowdown of ice retreat, broadly consistent with the abrupt 8.2 ka cooling event (Alley et al.,
       1997). The formation of the M1 moraine (8.6 ± 0.3 ka) could be related to a period of ice margin stagnation
       in response to local cooling at this time. Similarly, at Schuchert Dal in Scoresby Sund, radiocarbon dates
       of shells and their position in relation to the historical moraines suggest minimal ice sheet advance during
the 8.2 ka event (Denton et al., 2005). A possible explanation for ice margin stillstand or slowdown at both
       East Greenland sites is seasonality, with cooling primarily in the winter, similar to the Younger Dryas
       (Denton et al., 2005). In West Greenland, surface exposure dating of moraine systems record the ice margin
       responded directly to the 9.3 and 8.2 ka cooling events (Young et al., 2013b). These observations support
       regional cooling across Greenland during this interval. However, considering present-day Storstrømmen is
a surge glacier (Andersen et al., 2025; Mouginot et al., 2018; Reeh et al., 1994), an alternative explanation
       is that the moraine records past surge dynamics, similar to modern-day processes.



The response of the GrIS to climate variability is strongly influenced by ice flux (Young, et al., 2013b). In low ice flux glaciers, such as Storstrømmen, mass balance change driven by regional cooling (i.e., reduced summer ablation), may occur slowly resulting in a subtle response or stillstand. In contrast, the response to high flux glaciers, such as Jakobshavn Isbræ in West Greenland, to such cooling may occur relatively quickly resulting in an ice sheet advance (Young, et al., 2013b). For example, at Jakobshavn Isbræ, a high flux environment was observed between ~8.5 – 8.1 ka, directly depositing moraines during the 8.2 ka event (Young et al., 2013b). Our observations in northeast Greenland, are consistent with patterns observed in West Greenland (Young, et al., 2013b; Young et al., 2011), suggesting that marine-terminating glaciers show high sensitivity to abrupt warming and cooling on centennial timescales. In contrast, land-terminating glaciers in Greenland are less sensitive to abrupt climate change (Long et al., 2006; Young et al., 2011).

We estimate a minimum average retreat rate of ~28 m/yr for from the outer coast near Store Koldewey at ~12.7 ka (Larsen et al., 2022) to within ~3 km of the modern ice margin by ~8.6 ka. The average retreat rate of Storstrømmen during the last deglaciation is an order of magnitude lower than the observed grounding line retreat of 1.1 km at a rate of 250 – 380 m/yr between 2017 and 2021, attributed to surface ablation (Rignot et al., 2022). However, the modern retreat follows a surge event between 1978 and 1984 (Reeh et al., 1994), and present surge behavior may be related glacier instability. Therefore, direct comparisons between Holocene and modern retreat rates at Storstrømmen require caution. Additionally, differences in temporal resolution, the drainage system, and in external forcing such as ablation rates and ocean temperatures must be considered when comparing retreat rates across different timescales.



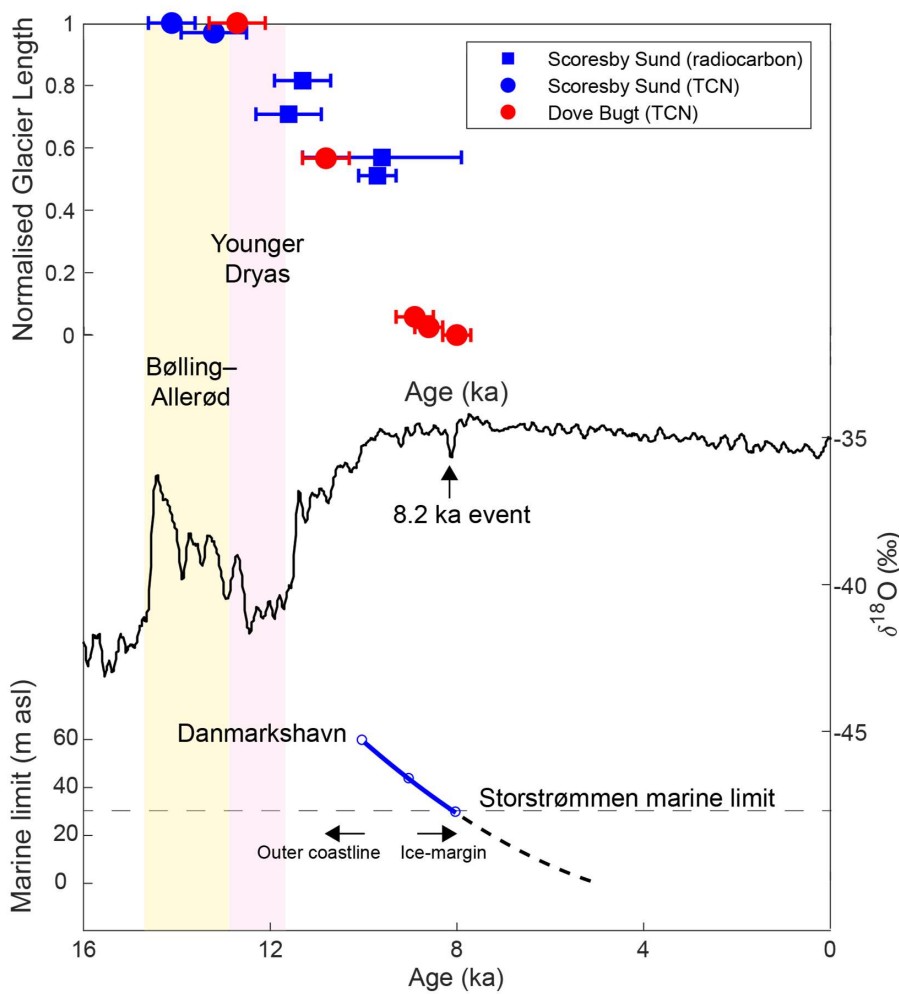

**Figure 7.** Ice margin retreat from Scoresby Sund and the Dove Bugt region. $\delta^{18}$O record of the North Greenland Ice core Project (NGRIP members, 2004). Marine limit curve for the Dove Bugt region derived from Weidick et al. (1996). Normalized glacier length includes published radiocarbon ages from Scoresby Sund (Dowdeswell et al., 1994; Funder, 1978; Marienfeld, 1991), and published $^{10}$Be ages from the Dove Bugt embayment (Larsen et al. 2022).

### 5.3 Rates of marine-terminating glacier retreat

As ice retreated westward out of the Greenland Sea, minimum average retreat rates of ice streams from the middle of the continental shelf to the outer coast of northeast Greenland were 96 m/yr from Foster Bugt to



the mouth of Kejser Franz Joseph Fjord (Olsen et al., 2022). At the Store Koldewey Trough, located west of Store Koldewey Island and the Dove Bugt embayment, retreat rates ranged between 80 – 400 m/yr (Olsen et al., 2020). In contrast, our minimum average retreat rates observed at Scoresby Sund (~43 m/yr) and Storstrømmen Glacier (~28 m/yr) are consistent with broader estimates for marine-terminating glaciers in

East Greenland fjords, which range between 10 and 80 m/yr (Dyke et al., 2014; Hughes et al., 2012).

We summarize the late glacial and Holocene rates of retreat in East Greenland in Figure 8. For comparison, retreat rates include 33 m/yr at Nordfjord, and 35 m/yr at Kejser Franz Joseph Fjord (Olsen et al., 2022); ~10 m/yr at Kangerlussuaq Fjord (Dyke et al., 2014); and ~80 m/yr at Helheim Glacier (Hughes et al., 2012). Rapid retreat rates of 79N Glacier within Nioghalvfjerdsfjorden after 10 ka range between 248.5

m/yr and 613 m/yr record some of the highest retreat rates in both Greenland and Antarctica (Roberts et al., 2024). Our retreat rates at Scoresby Sund (~43 m/yr) are comparable to retreat rates for the entire central East Greenland region (35 ± 23 m/yr), while our retreat rates at Dove Bugt (~28 m/yr) are lower than for the entire Northeast Greenland region (44 ± 10 m/yr) that were estimated using transects across isochrones (Leger et al., 2024). In West Greenland, Young et al. (2011) estimated an average retreat rate of ~100 m/yr

for Jakobshavn Isbræ between 8 – 7.5 ka. The thickest and fastest moving parts of Greenland's largest present-day ice streams correspond to the contemporary submarine melt rates at 79N (50 – 60 m/yr) in northeastern Greenland, and Petermann Glacier (40 – 50 m/yr) and Ryder Glacier (50 m/yr) in northwestern Greenland (Wilson et al., 2017). These modern rates are comparable to late glacial and Holocene retreat rates observed in East Greenland (Fig. 8).

Fast flowing glaciers that lack topographic pinning points that enhance ice stability (i.e., islands, upward sloping beds) are particularly susceptible to rapid retreat (Mouginot et al., 2015). Modern ice flow rates at Jakobshavn (11–17 km/yr; Joughin et al., 2014), Helheim (5–11 km/yr; Bevan et al., 2012) and 79N (1.4 km/yr; Vijay et al., 2019), all reflect fast flowing outlet glaciers of the GrIS with large volumes of ice discharge. By comparison, Storstrømmen Glacier has flow speeds of 0.19 km/yr (Vijay et al., 2019), an

order of magnitude slower than the fastest flowing glaciers of Greenland. This observation is also reflected in late glacial and Holocene retreat patterns in East Greenland fjords. With the exception of 79N Glacier, retreat rates observed within the East Greenland fjords during the late glacial and Holocene were generally lower than those observed from the middle shelf to the outer coast. This pattern suggests a shift from large ice streams to topographically controlled retreat within fjord systems.

Numerical simulations from southwestern Greenland suggested ice calving within fjord systems significantly impact enhanced ice discharge at the terminus of outlet glaciers during the early Holocene (Cuzzone et al., 2022). Therefore, future ice sheet modeling, benchmarked against geologic constraints of




past East GrIS change (e.g., Briner et al., 2020) could greatly improve estimations of ice mass loss, rates or retreat, and associated processes from marine-terminating outlet glaciers across Greenland.

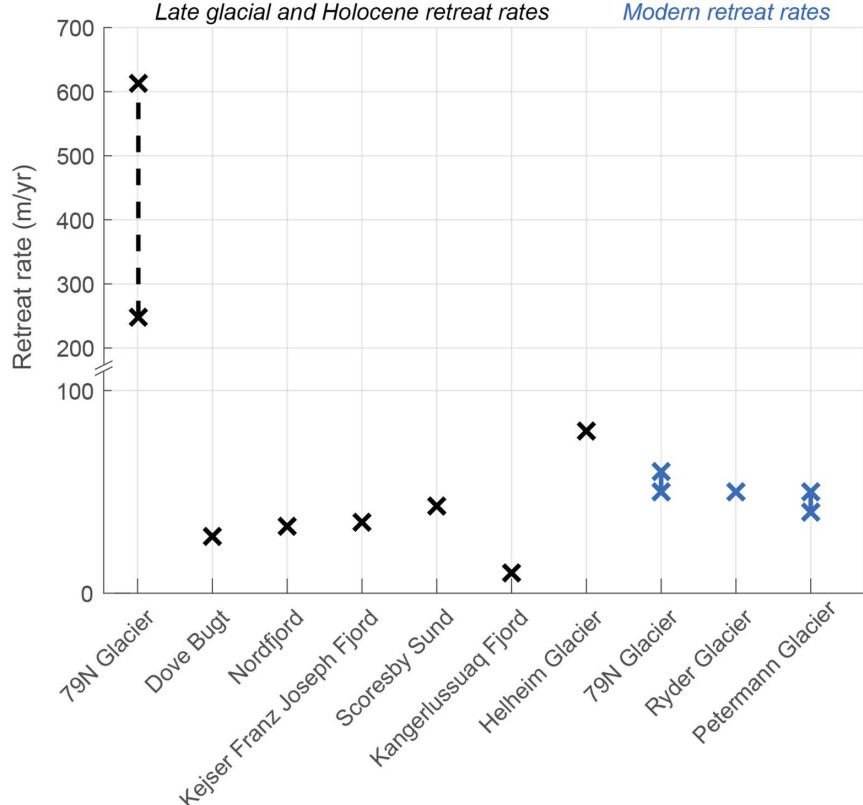


**Figure 8.** Rates of ice margin retreat (m/yr) for marine-terminating outlet glaciers in East Greenland during the late glacial and Holocene, compared with modern subglacial melt rates from 79N Glacier, Ryder Glacier and Petermann Glacier (Wilson et al., 2017).

**6 Conclusion**

We report new [10]Be ages from Scoresby Sund on the outer coast in central east Greenland, and from near the modern ice margin of Storstrømmen Glacier in northeast Greenland. These new exposure ages provide direct constraints of the late glacial and early Holocene timing of ice retreat westward out of the Greenland Sea, within the present-day coastline, and within the modern ice margin. Our key findings include:

- Ice retreated from Rathbone Island, east of Scoresby Sund, by ~14.1 ka, recording one of the earliest
known terrestrial deglaciation events in Greenland. Our [10]Be ages support the idea that ice margin



retreat across the East Greenland shelf was perhaps in response to enhanced sensitivity of abrupt global sea-level rise.

- Ice retreated from the mouth of Scoresby Sund at ~13.2 ka during the Bølling-Allerød warm period, perhaps related to an influx of warm Atlantic water that flowed onto the East Greenland continental shelf (Hansen et al., 2022).

- Storstrømmen Glacier reached ~3 km from the modern ice margin at ~8.6 ka and reached the modern margin ~8 ka. The ice margin slowed or reached a stillstand between 8.6 – 8.0 ka. In low ice flux environments, such as Storstrømmen, mass balance change is driven by regional cooling and may occur slowly resulting in a subtle response or stillstand. These observations suggest that marine-terminating sectors of the eastern GrIS may be highly sensitive to abrupt warming and cooling events on centennial timescales.

- Average rates of marine-terminating glacier retreat at Scoresby Sund were ~43 m/yr between ~13.2 ka and 9.7 ka. Average rates of marine-terminating glacier retreat in the Dove Bugt region were ~28 m/yr between ~12.7 ka and ~8.6 ka. The rates of late glacial and Holocene retreat at Scoresby Sund, and Storstrømmen Glacier are within previously reported estimates of marine-terminating outlet glacier retreat across eastern Greenland fjords (10 – 80 m/yr; Dyke et al., 2014; Hughes et al., 2012). These retreat rates are also comparable to modern retreat rates observed at the largest ice streams in northeastern (50 – 60 m/yr), and northwestern Greenland (40 – 50 m/yr; Wilson et al., 2017).

- The exposure ages from both regions of East Greenland provide robust boundary conditions for ice sheet models, improving the accuracy of simulating past GrIS change. These data refine our understanding of the timing, rates and mechanisms of ice sheet retreat during the last deglaciation, and provide important context for understanding contemporary rates of ice mass loss from the GrIS.

**Data availability.** All data described in the paper are included in the Supplementary files.

**Author contributions**

NEY, JPB and JMS designed the study. NEY, AB-K, KKP, CKW-G, BG, JPB and JMS conducted fieldwork and sample collection. JTHA and KKP processed the samples and JTHA conducted sample analysis and initial interpretations. JTHA prepared the manuscript with contributions from all authors.

**Competing interests**



This information product has been peer reviewed and approved for publication as a preprint by the U.S. Geological Survey.

## Acknowledgements

We thank Volcano Heli for field support; Liza Wilson for assistance in the field; and Roseanne Schwartz, Jean Hanley, Kylie Seward, and Maya Lasker for laboratory assistance. Alan Hidy and Tyler Anderson are thanked for AMS measurements at Lawrence Livermore National Laboratory. The USGS internal reviewers are thanked for their feedback. Any use of trade, firm, or product names is for descriptive purposes only and does not imply endorsement by the U.S. Government.

## Financial support

The research was supported by the National Science Foundation Office of Polar Programs (Awards #2105908 to NEY and JMS; #2106971 JPB).

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
