# Peer review of "East Greenland Ice Sheet retreat history from Scoresby Sund and Storstrømmen Glacier during the last deglaciation"

_EGUsphere, 2025_

## Author Comment (AC1)

Response to Anonymous Referee #1

We thank the referee for their comments and have addressed each comment below in red. Blue text is what is included in the revised version.

The manuscript by Anderson et al provides new information on the last deglaciation of NE Greenland. Based on 29 $^{10}$Be ages from Scoresby Sund and Storstrømmen Isbræ, they add to the existing geochronological data from the area. Overall, the new data is not surprising but generally supports the existing deglaciation chronology. The data is furthermore used to constrain the retreat rates from the outer coast to the present-day ice margin. Based on the compilation of data, they calculated retreat rates of 43 to 28 m/yr. These estimates are similar to what has previously been reported and identical to modern observations of ice retreat. One could be critical and argue that the new results lack novelty and are insufficient to warrant a new publication. However, I find it very valuable as it provides more data to constrain the deglaciation of NE Greenland – an area where little work has been done. Accordingly, I recommend publication as it provides one more piece in the puzzle of the deglaciation history of NE Greenland.

Overall, I find the manuscript well-written, and the data support the conclusions. Thank you. Besides some general comments, I have only a few minor comments that are all provided to improve the quality of the manuscript.

General comments:

- Re-calibration of $^{14}$ The regional dR (-46 +/- 57) for East Greenland (cf. Pearce et al., 2023) has not been used. Instead, a local dR has been used. However, it is unclear how the local dR's have been determined. Did you take the average of the nearest datapoints for each sample? Yes, we calculated the average dR. We have now reported this on line 249. Additionally, Supplementary Table 3 lists the number of data points at each individual site.

- The title is a little misleading. It gives the impression that the focus of the study is the regional ice sheet history of East Greenland. However, this is not the case, and the title should reflect that it is a more local study of two study sites in Scoresby Sund and Storstrømmen Isbræ. We have changed the title to: East Greenland Ice Sheet retreat history from Scoresby Sund and Storstrømmen Glacier during the last deglaciation

- The potential link between a readvance/stabilisation at Storstrømmen Glacier and the 8.2 ka cold event is rather weak and should be toned down (see comment below). We have deleted sentences on lines 384 – 388 about the potential connection to the 8.2 ka event (now lines 405 – 409). We have also

added that the M1 moraine could indeed be unrelated to a climate event. We have added this additional explanation on line 404 - 405: The formation of the M1 moraine (8.6 ± 0.3 ka) could be related to a period of ice margin stagnation in response to local cooling or localized dynamics unrelated to climate at this time.

We also state on lines 414 – 415: Another alternative explanation is that the ice margin was pinned to the island ~8.6 ka, slowing ice retreat during this time, independent of any climate event.

Minor comments:

Line 14: As written in the abstract – the study area concerns Scoresby Sund and Storstrømmen Glacier and this should be reflected in the title. We have changed the title. See comment above.

Line 45: Change Storstrømmen "Isstrøm" to "Glacier" This has been changed

Line 57: It would be relevant to add a sentence on the Holocene history of NEGIS i.e. Weidick et al., 1996; Bennike and Weidick 2001; Larsen et al., 2018; Roberts et al., 2024 We have added an additional sentence about Holocene history of NEGIS on lines 59 and 60: The NEGIS retreated from the outer coast to the present ice margin between 11.7 and ~9.0 ka (Larsen et al., 2018; Roberts et al., 2024).

Line 60: Leger et al have made an excellent review of the overall deglaciation history of Greenland based on existing data. However, the citation to their work here is misplaced as they don't provide any new data in East Greenland, unlike the other cited papers. This has been removed

Line 70: In Larsen et al., 2022, there are more comprehensive ice margin outlines in NE and E Greenland that could be added to the figure. We have added the Milne Land stade moraines to Fig. 1

Line 96: Roberts et al., 2024 are determining the thinning history of 79N based on several dip-stick profiles. For the retreat history, it would be more accurate to cite Bennike and Weidick 2001; Larsen et al., 2018. Thank you for bringing this to our attention. We have amended the text to include early Holocene retreat rates of Bennike and Weidick 2001; Larsen et al., 2018, and the lateral retreat rates calculated by Roberts et al., 2024. These changes are stated on lines 104 – 106: The 79N Glacier experienced slow early Holocene ice margin retreat from the outer coast to the present

ice margin (Larsen et al., 2018; Bennike and Weidick, 2001). However, retreat rates from the inner fjord suggest rapid deglaciation between 10 and 8.5 ka (Roberts et al., 2024).

Line 100-105: This paragraph is not very relevant for this study. We have deleted this paragraph.

Line 135: Add Germania Land and other place names to figure 1. The Milne Land moraines are discontinuous but have been inferred to run parallel to the coast from Scoresby Sund to Germania Land. However, it has only been dated around Scoresby Sund and the correlation is uncertain. We have included Germania Land and the Milne Land stade moraines to Fig. 1, and added "(Fig. 1)" on line 146.

Figure 3: The age estimate from Levy et al is based on several 10Be dates. I suggest adding this information to the figure caption or to the figure as n=x next to the 10Be age. We have added "n = 8" in the caption.

I would make Panel A bigger and add the weighted mean from each site below the individual 10Be dates. Then Panel B could be omitted. We have made Panel A bigger and added the weighted mean from Kap Brewster and Uunarteq. Panel B has been deleted.

Line 159:  Change Storstrømmen "Isstrøm" to "Glacier" This has been changed.

Figure 4: The age estimates are based on several 10Be dates from each site. I suggest adding this information to the figure caption or to the figure as n=x next to the 10Be age. We have included these values in the figure caption. It would also be relevant to include the 10Be ages from Håkansson et al. and Skov et al on the figure, although they are not used to calculate retreat rates. There are many 10Be and radiocarbon ages that could be included from this study area and figure. However, adding these would make the figure "crowded" and harder to follow. For simplicity and to stay focused on the retreat rates, we have decided not to include these. However, we have now highlighted these studies on lines 102 -104.

Line 230 (table 2): Why are all the samples from Storstrømmen used to calculate a mean age? It would be more relevant to provide the mean ages of M1, and outboard. We have updated Table 2 to show the mean ages from outboard, M1 moraine and inboard sites.

Line 321. In the discussion of the deglaciation of Scoresby Sund I miss a discussion of the previous work of Hall et al., 2008a, b; Kelly et al., 2008 from Kjove Land, Gurreholm dal and Schuchert dal. Including the new work by Kelly et al. (2025), we have added additional discussion stated on lines 357 – 359: This retreat chronology is consistent

with mountain glacier retreat between ~12.8 and 11.7 ka from Milne Land, Kjove Land, Gurreholm Dal and Schuchert Dal derived from $^{10}$Be and radiocarbon ages (Hall et al., 2008; 2010; Kelly et al., 2008; 2025; Levy et al., 2016).

Figure 6: I suggest adding the calculated weighted mean, excluding outliers from table 2, to the camel plots. We have added the weighted means, excluding outliers to Fig. 6

Figure 7: It is not clear which 10Be + 14C dates have been used to constrain the retreat history. Maybe this could be highlighted on Figure 3 + 4? We have added additional text in the captions of Fig. 3 stated on line 163: The grey dashed line denotes the flowline and $^{10}$Be and radiocarbon ages used to calculate average retreat rates.

And in the caption of Fig. 4 stated on line 210 and 211: The grey dashed line denotes the flowline used to calculate average retreat rates. The mean $^{10}$Be ages used to calculate average retreat rates are 12.7 ka, 10.8 ka, and 8.9 ka.

Line 375: Within uncertainty, the M1 moraine (8.6 +- 0.3 ka) could be related to the 8.2 ka event when excluding one outlier of 9.6 ka. However, in theory, it might also be related to the 9.3 ka event if the samples dated reflect the stabilisation of the moraine rather than the deposition of the moraine cf. Heyman et al., 2011. Or something unrelated to climate - the glacier is retreating to a pinning point (the island), and this could halt the ice retreat and give rise to the deposition of moraine M1. Given that the authors only have data from one moraine, it is a little far-fetched to speculate that the ice sheet reacted to the 8.2 ka event despite compelling evidence from W Greenland (c.f. Young et al.). I urge the authors to tone down the potential relation to the 8.2 ka event (paragraphs line 375-397).

We have deleted sentences on lines 384 – 388 about the potential connection to the 8.2 ka event (now lines 405 – 409). We have also added that the M1 moraine could indeed be unrelated to a climate event. We have added this additional explanation on lines 404 - 405: The formation of the M1 moraine (8.6 ± 0.3 ka) could be related to a period of ice margin stagnation in response to local cooling or localized dynamics unrelated to climate at this time.

We also state on lines 414 – 415: Another alternative explanation is that the ice margin was pinned to the island ~8.6 ka, slowing ice retreat during this time, independent of any climate event.

---

## Author Comment (AC2)

Response to Anonymous Referee #2

We thank the referee for their helpful comments and our responses are in red below. Blue text is what is included in the revised version.

Anderson et al. present a chronology of post-Last Glacial Maximum Greenland Ice Sheet retreat at Scoresby Sund and Storstrømmen Isbræ based on cosmogenic nuclide $^{10}Be$ exposure dating. The exposure ages from erratic boulders and bedrock from these regions, when compared to radiocarbon ages and other in situ $^{10}Be$-dated samples, are consistent with the timing and rates of retreat documented at other sites across east and northeast Greenland. The patterns inferred from calculated ice margin retreat rates are also consistent with modern day observations. The additional dates and constraints presented in this study contribute to a more comprehensive understanding of Greenland ice sheet dynamics, and I would recommend publication. Below is a short list of minor comments.

Minor comments

It may be beneficial to readers who are new to this topic to provide a very brief description of the use of in situ $^{10}Be$ along with citations (e.g., Nishiizumi et al., 1989; Lal, 1991; maybe Ivy-Ochs and Briner, 2014 for an overview/summary) so they can better familiarize themselves with the dating methods and other concepts (i.e., inheritance) discussed in this paper. We have added a sentence about the use of in situ cosmogenic nuclides with the suggested citations stated on lines 64 – 66: To record ice margin change, glacial landforms (such as erratic boulders and glacially polished bedrock) can be dated via in situ cosmogenic nuclides produced in rock surfaces (e.g. Ivy-Ochs and Briner, 2014; Lal, 1991; Nishiizumi et al., 1989).

L45: The acronym "SI" is not used elsewhere in the manuscript – would recommend removing it. On a related note, "ZI" is only used once in L92. Perhaps consider just using Zachariae Isstrom instead. These have been removed and edited.

L91-99 provides a list of already available retreat chronologies in east and northeast Greenland. In addition to the general explanation in L58-65, it may benefit the reader to explain the reasoning why additional dates from the two specific regions presented in this study are necessary. We have added additional reasoning stated on lines 68- 70: Here, we build on previous work studying the retreat history of marine-terminating glaciers in East Greenland, and present in situ cosmogenic $^{10}Be$ exposure ages from (i) the outer coast of Scoresby Sund, which improve age constraints at the mouth of the largest fjord system in Greenland, and (ii) near the modern ice margin of Storstrømmen Glacier, where few existing ages constrain East GrIS retreat.

Fig.3B: Consider including ages at Uunarteq and Kap Brewster, as well. We have modified Fig. 3 as suggested by Referee 1 and deleted panel 3B.

Table 2: 23DMH-CR1 is italicized in Table 2 as an outlier, but it is not italicized in Fig.5 This has now been italicized in Fig. 5

L279: 26.5 ± 0.6 ka should be 26.4 ± 0.6 ka. This has been corrected

L307: "and are inboard of the M1 moraine," and "On the northern margin of the island…" We have made these corrections.

Fig.7: It may be considerate to define any acronyms (i.e., TCN) in the caption for those readers who first browse a paper's figures to understand the main points before reading in detail. We have included the definition in the caption.

L397: Delete "from". This has been deleted.

L416: Change "At the Store Koldewey Trough, located west of Store Koldewey Island…" to "At the Store Koldewey Trough, located east of Store Koldewey Island…" This has been changed to "east".

---

## Author Comment (AC3)

We thank Kelly et al. for suggesting we consider their newly published and recalculated data during the revision of our manuscript.

In particular, Kelly et al. comment that our new dataset sets up a 'stratigraphic inversion' with their newly published dataset from inner Scoresby Sund, and remark that ice 'recession to inboard of these moraines occurred by at least ~14 ka' – implying that ice must have retreated from our field sites at the outer coast and the fjord mouth to Kelly et al. (2025) field sites prior to ~14 ka.

In this comment on our manuscript, Kelly et al. seem to be suggesting that their chronology is at odds with our new dataset showing the outer coast was deglaciating around 14.1 ± 0.3 ka at Rathbone Island and ice was on its way up Scoresby Sund, passing Kap Brewster, around 13.2 ± 0.7 ka.

Although Kelly et al. (2025) write in their manuscript that deglaciation of landscapes beyond their outer moraines in inner Scoresby Sund 'occurred by ~14.0 ka', their own data (7 ages from boulders outboard the outer MLS moraines at Holger Danskes Briller and Kjove Land, excluding one 71 ka outlier) average 13.0 ± 0.8 ka.

This age (13.0 ± 0.8 ka) is consistent with their other peak [10]Be ages for the outer moraines (13.4, 13.7, 12.2 ka, no error ranges reported) deposited by mountain glaciers, and ice sheet outlets. We note that these three age assignments average 13.1 ± 0.7 ka, and agree with their radiocarbon-based age assignments of ~12.8 cal ka BP, and 'shortly before 12.80 and 12.65 cal ka BP.'

Given the above, and our age of 13.2 ± 0.7 ka at Kap Brewster and Uunarteq, we find no evidence supporting a 'stratigraphic inversion' between these datasets. In fact, the Kelly et al. (2025) dataset and our new coastal [10]Be constraints presented here, combined, suggest that Scoresby Sund retreated rapidly after deglaciation of Rathbone Island at 14.1 ± 0.3 ka and Kap Brewster at 13.2 ± 0.7 ka.